# A Graph-Theoretical View of Space Folding via the Motzkin–Straus Framework

**Michal Lewandowski**[†] **Bernhard Heinzl**[†] **Roman J. Rainer**[†]
**Bernhard Nessler**[†] **Bernhard A. Moser**[†*]

[†] Software Competence Center Hagenberg (SCCH)
[*] Johannes Kepler University Linz (JKU)
{name.surname}@scch.at

## Abstract

Understanding the internal geometry of neural network representations remains an open challenge in deep learning research. Recent work has introduced a measure of space folding that quantifies how convex regions in input space map to non-convex, folded structures in activation space via straight-path induced walks of binarized activation patterns. In this paper, this space folding measure is linked to the classical Motzkin–Straus theorem through the graph Lagrangian of an interval graph constructed from such walks. This connection expresses a discrete, path-based geometric statistic as a continuous quadratic objective, suggesting that space folding can be incorporated as a differentiable regularization term in gradient-based training to guide networks toward more compact internal representations.

## 1 Introduction

Binarized activation patterns induce a discrete state space for a neural network, where each input maps to a vertex of a binary hypercube and the Hamming distance measures how many neurons change state. Recent work proposed *space folding* as a path-based statistic on this discrete space, designed to quantify deviations from discrete convexity along straight input segments, which was related to a higher generalization capacity in downstream tasks (Lewandowski et al., 2025a). Notably, it was shown that higher aggregated space folding with the same performance of the networks (as measured by, e.g., classification accuracy), was shown to empirically correlate with more compact neural representations. Despite being easy to define, the resulting functional is piecewise-constant in the underlying network parameters which hinders its use as an optimization criterion with gradient-based methods (Lewandowski et al., 2025b). In this paper, we provide a graph-theoretic representation of space folding for straight-path-induced activation walks. Given a refined unit-flip walk $\Gamma$ in $\{0, 1\}^N$, we build an induced *interval graph* (i.e., a graph whose vertices correspond to intervals on the real line and whose edges connect pairs of overlapping intervals) $G^*$ from mismatch blocks of coordinates relative to the start pattern. The key observation is that the maximum Hamming distance along the walk equals the maximum overlap of these mismatch intervals, which is exactly the clique number $\omega(G^*)$. Using the Motzkin–Straus theorem (Motzkin & Straus, 1965), we rewrite space folding as an explicit transform of the graph Lagrangian $\Lambda(G^*)$. Our contributions are as follows. (i) We construct an interval graph $G^*$ induced by mismatch blocks of a refined activation walk. (ii) We algebraically link the space folding measure to the Motzkin–Straus Lagrangian. (iii) We give an exact algorithm to compute $\omega(G^*)$ and hence $\Lambda(G^*)$ from the walk.

## 2 Related Work

**Activation Space of Neural Networks.** The study of space partitions created by internal activation states of the model dates back to Makhoul et al. (1989). While its definition is straightforward for MLPs (Makhoul et al., 1991; Montúfar et al., 2014), its study in larger models is more involved, see e.g. Balestriero et al. (2024). The created partition has been used, among others, to gauge the expressivity of neural networks: the finer the partition the more expressive the network (Montúfar et al., 2014; Raghu et al., 2017; Serra et al., 2018; Hanin & Rolnick, 2019). This stems from the fact

that each element of partition is a region where the behavior of the model is locally constant, akin to the concept of a basis in algebra or analysis. A recently introduced space folding measure, operating on binarized activations, is constructed to quantify *deviations* from convexity in the discrete sense (Lewandowski et al., 2025a; 2026).

**Graph Theory and Neural Networks.** Graph theory provides a natural language for describing pairwise relations and global invariants in structured systems. Graph Neural Networks (GNNs) have emerged as a powerful framework for exploiting relational structure in data (Scarselli et al., 2009; Hamilton et al., 2017; Veličković et al., 2018; Xu et al., 2019). From a different perspective, recent work has linked fully connected ReLU networks to connectivity graphs whose nodes represent linear regions and whose edges connect regions sharing a face (Gaines & Bi, 2026). Related preliminary work defines a graph in which nodes correspond to activation-induced linear regions and edges join regions that differ by a single neuron flip (Dhayalkar, 2025). Our approach differs in that we construct a graph from the temporal overlap of mismatch blocks, treating each block as a vertex.

## 3 PRELIMINARIES

**Space Folding.** For a model (e.g., a neural network) $\mathcal{N} : \mathcal{X} \to \mathcal{Y}$ consider two data points $\mathbf{x}_1, \mathbf{x}_2 \in \mathcal{X}$ in the (Euclidean) input space and the straight line between them, $[\mathbf{x}_1, \mathbf{x}_2] := (1 - t)\mathbf{x}_1 + t\mathbf{x}_2$, with intermediate points realized by varying the parameter $t \in [0, 1]$. To obtain a walk through activation patterns, we map $[\mathbf{x}_1, \mathbf{x}_2]$ through $\mathcal{N}$ to a *path* $\Gamma := \{\pi_1, \dots, \pi_n\} \in \mathcal{H}^{n \cdot N}$ in the (Hamming) activation space, where the intermediate activation patterns belong to a binary hypercube, $\pi_i \in \mathcal{H}^N$, $\forall i$. The space folding measure is defined (cf. Lewandowski et al. (2025a; 2026)) as the ratio of two range measures $r_1, r_2 : \Gamma \to \mathbb{R}$ subtracted from one, i.e.,

$$\chi(\Gamma) := 1 - \frac{r_1(\Gamma)}{r_2(\Gamma)} = 1 - \frac{\max_i d_H(\pi_i, \pi_1)}{\sum_{i=1}^{n-1} d_H(\pi_i, \pi_{i+1})} \in [0, 1], \tag{1}$$

where $r_1(\Gamma) := \max_i d_H(\pi_i, \pi_1)$ and $r_2(\Gamma) := \sum_{i=1}^{n-1} d_H(\pi_i, \pi_{i+1}) > 0$ (more details in App. A).

**A Primer on Graph Theory.** We now gather the necessary notions from graph theory. For a more systematic overview, we refer the reader to manuscripts such as Diestel (2017). We start with a definition of a *clique* and *clique number*.

**Definition 3.1** (Clique Number)**.** *Let $G = (V, E)$ be an undirected graph with vertex set $V = \{1, \dots, n\}$ and edge set $E \subseteq \{\{i, j\} : 1 \le i < j \le n\}$. A* clique *is a subset $C \subseteq V$ such that every distinct pair in $C$ forms an edge. The* clique number $\omega(G)$ *is the maximum size of a clique in $G$.*

A large clique is a large set of vertices that can all co-occur under the adjacency relation given by $E$. Computing the clique number $\omega(G)$ is NP-complete for general graphs, and the Motzkin–Straus objective motivates continuous optimization heuristics and relaxations for clique-like problems (Bomze, 1997).

Another classical notion in Graph Theory is the *graph Lagrangian*, which associates to a graph $G$ the quadratic form $L_G$ optimized over the probability simplex $\Delta = \{x \ge 0 : \sum_i x_i = 1\}$.

**Definition 3.2** (Graph Lagrangian)**.** *Let $A \in \{0, 1\}^{n \times n}$ be the* adjacency matrix *of G, i.e. $A_{ij} = 1$ if $\{i, j\} \in E$ and $A_{ij} = 0$ otherwise. Put $A_{ii} = 0$, and define the simplex*

$$\Delta := \left\{ x \in \mathbb{R}^n : x_i \ge 0 \,\forall i, \, \sum_{i=1}^{n} x_i = 1 \right\}.$$

*The* graph Lagrangian *of $G$ is defined as*[1] $\Lambda(G) := \max_{x \in \Delta} x^\top A x.$

The objective $x^\top A x$ is a (continuous) relaxation of clique structure, i.e., if $x$ is supported on a set of vertices that is nearly a clique, then many products $x_i x_j$ contribute through edges $\{i, j\} \in E$, which was used by Motzkin and Straus to give a new proof of Turán's theorem (Motzkin & Straus, 1965).

---

[1]Some authors define $\lambda(G) := \max_{x \in \Delta} \sum_{\{i,j\} \in E} x_i x_j$. For undirected graphs, $\Lambda(G) = 2\,\lambda(G)$.

**Theorem 3.3** (Motzkin–Straus). *Let $G$ be a simple undirected graph. Then[2]*

$$\Lambda(G) \;=\; 1 - \frac{1}{\omega(G)}.$$

*Moreover, the maximum is attained by taking $x$ uniform on a maximum clique: if $C \subseteq V$ is a clique with $|C| = \omega(G)$ and*

$$x_i = \begin{cases} 1/\omega(G), & i \in C, \\ 0, & i \notin C, \end{cases}$$

*then $x \in \Delta$ and $x^\top A x = 1 - 1/\omega(G)$.*

Theorem 3.3 bridges a discrete invariant $\omega(G)$ and a continuous optimization problem $\Lambda(G)$. In the next section, we build a graph from the unit interval graph in the discrete activation space and use it to rewrite the space folding functional as an explicit function of $\Lambda(G^*)$.

## 4    Graph-Lagrangian Representation of Folding

In this section we prove a graph Lagrangian equivalence of the space folding measure as defined in Eq. (1). We start by showing that we can refine a path $\Gamma$ such that all flips that decrease $r_1$ occur first followed by the flips that increase it. We denote the refined path with $\widetilde{\Gamma}$. Such a refinement does not affect the folding measure, which we phrase as Lemma 4.1 (see App. B for its complete proof).

**Lemma 4.1** (Refinement preserves $r_1$ and $r_2$). *Let $\Gamma = (\pi_1, \ldots, \pi_n) \subset \{0,1\}^N$ and refine each transition $\pi_i \to \pi_{i+1}$ into a unit-flip subsequence by flipping each differing coordinate exactly once. Fix the start pattern $\pi_1$. For each transition, partition the flipped coordinates into*

$$D_i := \{j : \pi_i[j] \neq \pi_1[j] \text{ and } \pi_{i+1}[j] = \pi_1[j]\}^3 \quad \text{(distance decreases)}$$

$$I_i := \{j : \pi_i[j] = \pi_1[j] \text{ and } \pi_{i+1}[j] \neq \pi_1[j]\} \quad \text{(distance increases)}.$$

*Define the refined walk $\widetilde{\Gamma}$ by ordering the flips within each transition so that all flips in $D_i$ occur first, followed by all flips in $I_i$ (in any order). Then (i) $\max_{v \in \widetilde{\Gamma}} d_H(\pi_1, v) = \max_i d_H(\pi_1, \pi_i) = r_1(\Gamma)$, and (ii) the number of unit flips in $\widetilde{\Gamma}$ equals $r_2(\Gamma)$, hence $\chi(\Gamma) = \chi(\widetilde{\Gamma})$ whenever $r_2(\Gamma) > 0$.*

We illustrate the idea of Lemma 4.1 in Example 4.1.

**Example 4.1.** *Let $\pi_1 = (0000), \pi_2 = (1100), \pi_3 = (0011)$ and consider the walk $\Gamma = (\pi_1, \pi_2, \pi_3)$. Then, $d_H(\pi_1, \pi_2) = 2 = d_H(\pi_1, \pi_3)$, hence $r_1(\Gamma) = 2$, and $r_2(\Gamma) = d_H(\pi_1, \pi_2) + d_H(\pi_2, \pi_3) = 6$. A unit-flip refinement of $\pi_2 \to \pi_3$ that flips the* increasing *bits first is:*

$$(1100) \to (1110) \to (1111) \to (0111) \to (0011).$$

*Its distances to $\pi_1 = (0000)$ are:*

$$2 \to 3 \to 4 \to 3 \to 2,$$

*so for the refined walk $r_1(\widetilde{\Gamma}) = 4 \neq r_1(\Gamma)$. Instead, in Lemma 4.1 we propose to refine the walk as follows. For transition $\pi_2 \to \pi_3$ w.r.t. initial $\pi_1$: bits $1, 2$ change $1 \to 0$ (distance* decreases *w.r.t. $\pi_1$), and bits $3, 4$ change $0 \to 1$ (distance* increases *w.r.t. $\pi_1$). Flipping all decreases first, then increases, yields:*

$$(1100) \to (0100) \to (0000) \to (0001) \to (0011),$$

*with distances:*

$$2 \to 1 \to 0 \to 1 \to 2.$$

*Therefore it holds that $\max_{v \in \widetilde{\Gamma}} d_H(\pi_1, v) = 2 = r_1(\Gamma)$, while the total number of unit flips is unchanged, i.e., $r_2(\widetilde{\Gamma}) = r_2(\Gamma) = 6$.*

We are now ready to state and prove our main result.

---

[2]Under the alternative normalization, $\lambda(G) = \frac{1}{2}(1 - 1/\omega(G))$.

[3]$\pi_i[j]$ denotes the $j^{\text{th}}$ coordinate of $\pi_i$, e.g., if $\pi_i = (101)$ then $\pi_i[1] = 1, \pi_i[2] = 0, \pi_i[3] = 1$.

**Theorem 4.2.** *Let $\Gamma$ be a walk on $\{0,1\}^N$ and let $\widetilde{\Gamma}$ be the refined walk from Lemma 4.1 of length $L = r_2(\Gamma)$. Then there exists a graph $G^*$ such that*

$$\chi(\Gamma) = 1 - \frac{1}{L\,[1 - \Lambda(G^*)]},$$

*where $\Lambda(G^*)$ is the Motzkin-Straus graph Lagrangian.*

*Proof.* We work with the refined unit-flip walk $\widetilde{\Gamma} = (\widetilde{\pi}_1, \ldots, \widetilde{\pi}_{L+1})$. For each coordinate (neuron) $j \in \{1, \ldots, N\}$ define the mismatch as $m_j(t) := \mathbf{1}\{\widetilde{\pi}_t[j] \neq \widetilde{\pi}_1[j]\}$, $t \in \{1, \ldots, L+1\}$. By definition of the Hamming distance, $d_H(\widetilde{\pi}_1, \widetilde{\pi}_t) = \sum_{j=1}^{N} m_j(t)$.

**Step 1: Each mismatch set is a finite union of disjoint blocks.** Fix $j$. Let $1 \leq \tau_{j,1} < \tau_{j,2} < \cdots < \tau_{j,k_j} \leq L$ be the (finite) set of times at which bit $j$ flips between $\widetilde{\pi}_\tau$ and $\widetilde{\pi}_{\tau+1}$. Between two consecutive flip times, the value of bit $j$ is constant, hence $m_j(t)$ is constant. Therefore the set

$$S_j := \{t \in \{1, \ldots, L+1\} : m_j(t) = 1\}$$

is a finite union of maximal disjoint integer intervals.

**Step 2: Build an interval graph whose clique number equals maximum overlap.** Create one vertex for each maximal block in $S_j$ (this is the vertex-splitting step), across all $j$. Identify each vertex with its time-interval (block). Define $G^*$ as the intersection graph of these intervals: two vertices are adjacent iff their blocks overlap in time. For any time index $t$, each neuron $j$ contributes *exactly one* active block iff $m_j(t) = 1$. Hence the number of interval-vertices covering $t$ equals

$$\#\{\text{interval-vertices covering } t\} = \sum_{j=1}^{N} m_j(t) = d_H(\widetilde{\pi}_1, \widetilde{\pi}_t).$$

Because each vertex $v$ corresponds to a (discrete) interval $I_v \subseteq \{1, \ldots, L+1\}$ on a line, $G^*$ is an interval graph (Golumbic, 1980). Moreover, if $\{I_v : v \in C\}$ is a finite family of intervals that intersects pairwise (i.e., $C$ is a clique), then all intervals share a common intersection time. Indeed, write each $I_v = [\ell_v, r_v]$. Let $\ell = \max_{v \in C} \ell_v$ and $r = \min_{v \in C} r_v$. If $\ell > r$, pick $u$ with $\ell_u = \ell$ and $w$ with $r_w = r$; then $I_u \cap I_w = \emptyset$, contradicting pairwise intersection. Hence $\ell \leq r$ and any $t \in [\ell, r]$ belongs to every $I_v$ (we used here the so-called Helly property, see Golumbic (1980)). Thus every clique corresponds to a set of intervals covering a common time $t$, and conversely all intervals covering a fixed $t$ form a clique. Therefore,

$$\omega(G^*) = \max_t \#\{\text{interval-vertices covering } t\} = \max_t d_H(\widetilde{\pi}_1, \widetilde{\pi}_t).$$

By Lemma 4.1, $\max_t d_H(\widetilde{\pi}_1, \widetilde{\pi}_t) = r_1(\Gamma)$, and $L = r_2(\Gamma)$. Consequently,

$$\chi(\Gamma) = 1 - \frac{r_1(\Gamma)}{r_2(\Gamma)} = 1 - \frac{\omega(G^*)}{L}. \tag{2}$$

**Step 3: Apply Motzkin-Straus.** Let $A^*$ be the adjacency matrix of $G^*$ and define the graph Lagrangian

$$\Lambda(G^*) := \max_{x \in \Delta} x^\top A^* x, \qquad \Delta := \{x \geq 0, \ \mathbf{1}^\top x = 1\}.$$

By Motzkin–Straus, $\Lambda(G^*) = 1 - \frac{1}{\omega(G^*)} \Leftrightarrow \omega(G^*) = \frac{1}{1 - \Lambda(G^*)}$. Substituting this into Eq. (2) yields

$$\chi(\Gamma) = 1 - \frac{1}{L\,[1 - \Lambda(G^*)]}.$$

$\square$

We provide the algorithm and the analysis of its complexity in Appendix C. We illustrate the idea of the the proof of Theorem 4.2 in Example 4.2.

Table 1: Activation patterns $\pi_i$ used in Example 4.2.

| $\pi_1$ | $\pi_2$ | $\pi_3$ | $\pi_4$ | $\pi_5$ | $\pi_6$ | $\pi_7$ |
|---|---|---|---|---|---|---|
| (000) | (100) | (110) | (010) | (011) | (001) | (000) |

**Example 4.2.** *Let $N = 3$ and consider the refined unit-flip path $\Gamma = \{\pi_1, \ldots, \pi_7\}$ as given in Table 1. For each bit $j \in \{1, 2, 3\}$, recall that $m_j(t) = \mathbf{1}\{\pi_t[j] \neq \pi_1[j]\}$ for $t = 1, \ldots, 7$. Then*

$$m_1 : 0, 1, 1, 0, 0, 0, 0; \quad m_2 : 0, 0, 1, 1, 1, 0, 0; \quad m_3 : 0, 0, 0, 0, 1, 1, 0$$

*Thus, the Hamming distances to the start satisfy $d_H(\pi_t, \pi_1) = \sum_{j=1}^{3} m_j(t) = (0, 1, 2, 1, 2, 1, 0)$, so $r_1 = \max_t d_H(\pi_t, \pi_1) = 2$, $r_2 = L = 6$, and $\chi = 1 - r_1/r_2 = 2/3$. Each $m_j$ has 1's on a the following positions: $[2, 3]$, $[3, 5]$, $[5, 6]$, in discrete intervals. The interval graph on these runs has clique number $\omega = 2$, which equals $r_1$, hence $\chi = 1 - \omega/L$.*

**Remark 4.3.** *We remark that the construction introduced in Lemma 4.1 may not be unique. Different valid orderings could produce different graphs $G^*$. By Theorem 4.2, however, all such graphs share the same clique number $\omega(G^*)$ and therefore the same Motzkin-Straus value $\Lambda(G^*)$, ensuring that the stated equivalence for the space-folding measure holds regardless of the specific refinement chosen. In this work we focus on one such construction that suffices to establish the stated equivalence, and we leave a systematic study of how alternative refinements affect the induced graphs and their properties for future work.*

## 5    DISCUSSION

Our results provide a principled link between the discrete, path-based space folding measure and a continuous quadratic objective given by the Motzkin–Straus graph Lagrangian, thereby suggesting new ways to incorporate folding into gradient-based optimization. In particular, viewing folding through $\Lambda(G^*)$ opens the possibility of designing differentiable surrogates that approximate the original measure and can be used as regularization terms during training, guiding networks to develop more compact internal representations, in line with empirical evidence that increased folding correlates with better generalization and more efficient representations (Lewandowski et al., 2025a; 2026). Beyond direct regularization, folding-based objectives could be employed for architecture search (e.g., selecting models that exhibit desirable folding profiles), for monitoring training dynamics and detecting overfitting, or for probing robustness by comparing folding patterns along adversarial versus natural input paths (Lewandowski et al., 2026).

## 6    CONCLUSIONS AND FUTURE WORK

Our analysis shows that the notion of space folding can be related to the Motzkin-Straus graph Lagrangian of an interval graph constructed from mismatch blocks, offering a potential bridge between geometric and graph-theoretic descriptions of neural representations. This theoretical connection not only interprets folding as maximum temporal overlap of activation mismatches but also equips it with a closed-form continuous expression suitable for analysis and approximation. While the current formulation assumes binarized activations and unit-flip refinements along linear input paths, it lays a foundation for extensions to global folding statistics across multiple paths, potentially enabling new tools for characterizing the complexity and compactness of neural representations at scale.

## ACKNOWLEDGEMENTS

The research reported in this paper was partially funded by the Austrian Research Promotion Agency (FFG) through the project *AI4SustainablePT* (No. 913618); by the State of Upper Austria within the framework of the SCCH competence center INTEGRATE (FFG Grant No. 892418); by BMK, BMAW, BMIMI, and BMWET through the FFG COMET Competence Centers for Excellent Technologies Program; and by the Upper Austria business and research strategy *#upperVISION2030* within the framework of the AI Engineering and Certification Center (No. Wi-2022-699557-Hub).

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

## A CONSTRUCTION OF THE SPACE FOLDING MEASURE

Consider a straight line connecting two input points $\mathbf{x}_1, \mathbf{x}_2$ in the Euclidean input space. The intermediate points are realized by varying the parameter $t$ in a convex combination $(1-t)\mathbf{x}_1 + t\mathbf{x}_2$. For a practical implementation, Lewandowski et al. (2025a) spaced the parameter $t$ equidistantly on $[0,1]$, creating $n$ segments. Equal spacing, though easy and fast to implement, frequently results in suboptimal choice of the intermediate points. To obtain a walk through activation patterns, we map the straight line $[\mathbf{x}_1, \mathbf{x}_2]$ through a neural network $\mathcal{N}$ to a *path* $\Gamma := (\pi_1, \ldots, \pi_n) \in \mathcal{H}^{n \cdot N}$ in the Hamming activation space, where the intermediate activation patterns belong to a binary hypercube, $\pi_i \in \mathcal{H}^N$ for all $i \in \{1, \ldots, n\}$. We consider a change in the Hamming distance with respect to the initial activation pattern $\pi_1$ at each step $i$, $\Delta_i := d_H(\pi_{i+1}, \pi_1) - d_H(\pi_i, \pi_1)$, and then look at the maximum of the cumulative change $\max_k \sum_{i=1}^{k} \Delta_i$ along $\Gamma$,

$$r_1(\Gamma) = \max_i \sum_{j=1}^{i} \Delta_j = \max_i d_H(\pi_i, \pi_1). \tag{3}$$

We further keep track of the total distance traveled on the hypercube when following the path,

$$r_2(\Gamma) = \sum_{i=1}^{n-1} d_H(\pi_i, \pi_{i+1}). \tag{4}$$

For a measure of *space flatness*, we consider the ratio $r_1(\Gamma)/r_2(\Gamma)$. Equivalently, the *space folding* measure results as

$$\chi(\Gamma) := 1 - \max_i d_H(\pi_i, \pi_1) / \sum_{i=1}^{n-1} d_H(\pi_i, \pi_{i+1}). \tag{5}$$

The folding measure is lower and upper bounded, $\chi \in [0,1]$ (Lewandowski et al., 2025a).

## B PROOF OF LEMMA 4.1

**Lemma B.1** (Refinement preserves $r_1$ and $r_2$)**.** *Let $\Gamma = (\pi_1, \ldots, \pi_n) \subset \{0,1\}^N$ and refine each transition $\pi_i \to \pi_{i+1}$ into a unit-flip subsequence by flipping each differing coordinate exactly once. Fix the start pattern $\pi_1$. For each transition, partition the flipped coordinates into*

$$D_i := \{j : \pi_i[j] \neq \pi_1[j] \text{ and } \pi_{i+1}[j] = \pi_1[j]\} \quad \text{(distance decreases)}$$
$$I_i := \{j : \pi_i[j] = \pi_1[j] \text{ and } \pi_{i+1}[j] \neq \pi_1[j]\} \quad \text{(distance increases)}.$$

*Define the refined walk $\tilde{\Gamma}$ by ordering the flips within each transition so that all flips in $D_i$ occur first (in any order), followed by all flips in $I_i$ (in any order). Then (i) $\max_{v \in \tilde{\Gamma}} d_H(\pi_1, v) = \max_{1 \leq i \leq n} d_H(\pi_1, \pi_i) = r_1(\Gamma)$, and (ii) the number of unit flips in $\tilde{\Gamma}$ equals $r_2(\Gamma)$, hence $\chi(\Gamma) = \chi(\tilde{\Gamma})$ whenever $r_2(\Gamma) > 0$.*

*Proof.* Fix a transition $\pi_i \to \pi_{i+1}$ and let $d := d_H(\pi_1, \pi_i)$. Each unit flip changes $d_H(\pi_1, \cdot)$ by exactly $\pm 1$: flipping $j \in D_i$ decreases the distance by 1, while flipping $j \in I_i$ increases it by 1. Let $k_\downarrow := |D_i|$ and $k_\uparrow := |I_i|$. Under the prescribed ordering, the distance profile along the refined subpath is

$$d, \ d-1, \ \ldots, \ d-k_\downarrow, \ d-k_\downarrow+1, \ \ldots, \ d-k_\downarrow+k_\uparrow.$$

The final value equals $d_H(\pi_1, \pi_{i+1}) = d - k_\downarrow + k_\uparrow$. The maximum value attained on this subpath is therefore $\max\{d_H(\pi_1, \pi_i), d_H(\pi_1, \pi_{i+1})\}$ (all interior values lie between these). Taking the maximum over all transitions yields $\max_{v \in \tilde{\Gamma}} d_H(\pi_1, v) = \max_i d_H(\pi_1, \pi_i) = r_1(\Gamma)$. Finally, refinement does not change how many coordinates differ per transition: The refined subpath for $\pi_i \to \pi_{i+1}$ has exactly $d_H(\pi_i, \pi_{i+1})$ unit flips, so summing over $i$ gives $|\tilde{\Gamma}| - 1 = \sum_{i=1}^{n-1} d_H(\pi_i, \pi_{i+1}) = r_2(\Gamma)$. $\square$

## C CONSTRUCTION OF THE GRAPH LAGRANGIAN

In this section we provide an exact procedure (Algorithm 1) for construction of the graph Lagrangian for a given refined path-walk.

---

**Algorithm 1** Exact computation of $\Lambda(G^*)$ from a refined unit-flip walk

---

    **Input:** Refined unit-flip walk $\Gamma = (\pi_1, \ldots, \pi_{L+1}) \subset \{0,1\}^N$
    **Output:** $\omega(G^*), \Lambda(G^*)$
1:  $\mathcal{E} \leftarrow [\,]$                                    ▷ list of pairs (time, $\pm 1$); duplicates allowed
2:  $s[1..N] \leftarrow 0$                     ▷ $s[j] = 1$ iff coordinate $j$ currently mismatches $\pi_1$
3:  $a[1..N] \leftarrow$ None           ▷ $a[j]$ = start time of current mismatch block, if any
4:  **for** $t = 2$ to $L+1$ **do**
5:     $j \leftarrow$ FLIPPEDINDEX$(\pi_{t-1}, \pi_t)$
6:     $s[j] \leftarrow 1 - s[j]$
7:     **if** $s[j] = 1$ **then**
8:         $a[j] \leftarrow t$
9:     **else**
10:        $\mathcal{E} \leftarrow \mathcal{E} \cup \{(a[j], +1), (t, -1)\}$                ▷ block $[a[j], t-1]$
11:        $a[j] \leftarrow$ None
12:     **end if**
13: **end for**
14: **for** $j$ with $s[j] = 1$ **do**             ▷ close blocks still open at $t = L+1$
15:     $\mathcal{E} \leftarrow \mathcal{E} \cup \{(a[j], +1), (L+2, -1)\}$
16: **end for**
17: Sort $\mathcal{E}$ by time; ties: $(+1)$ before $(-1)$
18: $c \leftarrow 0, \omega \leftarrow 0$
19: **for all** $(\tau, \delta)$ in sorted $\mathcal{E}$ **do**
20:     $c \leftarrow c + \delta; \quad \omega \leftarrow \max(\omega, c)$
21: **end for**
22: $\Lambda(G^*) \leftarrow 1 - \frac{1}{\omega}$                           ▷ Motzkin–Straus
23: **return** $\omega, \Lambda(G^*)$

---

**Complexity.** Let $L$ be the number of unit flips in the refined walk $\Gamma$ and let $B$ denote the total number of mismatch blocks (equivalently, the number of interval-vertices in $G^*$). The algorithm generates exactly $2B$ endpoint updates (one $+1$ at each block start and one $-1$ after each block end). Constructing these updates takes $O(L)$ time, assuming the flipped coordinate index can be read in $O(1)$ per step (as holds for a unit-flip walk); closing any blocks still open at the end costs an additional $O(N)$. Sorting the (at most) $2B$ endpoint updates costs $O(B \log B)$, and the subsequent sweep that accumulates the overlap count costs $O(B)$. The total running time is therefore $T = O\big((L+N) + B \log B\big)$. Since each unit flip can start or end at most one mismatch block, we have $B \leq L$, hence

$$T = O\big(N + L \log L\big).$$

The memory usage is $O(N + B)$ for the per-coordinate state and the stored updates, i.e. $O(N + L)$ in the worst case.

**Remark C.1** (Direct $O(L)$ computation). *For a refined unit-flip walk $\tilde{\Gamma}$, we compute $r_1(\Gamma) = \max_t d_H(\tilde{\pi}_1, \tilde{\pi}_t)$ by a single scan: maintain $d_t = d_H(\tilde{\pi}_1, \tilde{\pi}_t)$ and update it by $\pm 1$ at each flip, then take $\max_t d_t$. Hence $\chi(\Gamma) = 1 - r_1/r_2$ is $O(L)$. We use the interval-graph view to make the maximum-overlap interpretation explicit and to support extensions where one obtains weighted overlap graphs or, in higher-dimensions, intersection graphs that are generally no longer interval, and thus relaxations via $\Lambda(G)$ become useful.*

