# OpenReview forum: "A Graph-Theoretical View of Space Folding via the Motzkin–Straus Framework"
_ICLR.cc/2026/Workshop/GRaM — ICLR 2026 Workshop GRaM Poster_

### Official Review · Reviewer_EYLg · 2026-02-17
**An interesting relationship between discrete and continuous quantities in understanding internal geometry of models**

**Rating:** 6
**Confidence:** 3

**Review:**

**Summary:**
The paper shows new theoretical results in space folding. Space folding being a measure to quantify how convex features map to non-convex regions through a neural network. This measure consists of an aggregation of hamming distances between binary outputs of neurons.The authors use the Motzkin-Straus theorem to link the original discrete measure to a continuous function, the graph lagrangian, thus opening the door to continuous relaxations of the original measure.

**Strengths:**
- The paper is technically sound and a good read.
- It proves an equivalence between a discrete quantity and a continuously defined one, opening the door to continuous relaxations of the problem that may enable richer ways to define the samples in t*x1+(1-t)x2.
- The proof relating the result with Motzkin-Straus is interesting.

**Weaknesses:**
- Lack of immediate impact or use-case of the findings.
- The measure still depends on the initial discrete formulation through L.
- Should highlight more the possible implications of the findings.

**Overall assessment:**
As a tiny paper I believe this should be interesting to the community, as the continuous relaxation could prove to be useful in further developments of space folding.

**Pmlr Suitability:**

NA

---

### Official Review · Reviewer_qrix · 2026-02-20
**Conceptually interesting graph-theoretic perspective on activation geometry, with open questions about relevance**

**Rating:** 6
**Confidence:** 2

**Review:**

**Summary**: This paper gives a graph-theoretic reformulation of a recently proposed “space folding” measure for neural network activations, which tracks how binarized activation patterns change along straight input segments. The authors show that the maximum Hamming distance along such a path can be expressed as the clique number of an induced interval graph, and then use the Motzkin–Straus theorem to rewrite the folding measure in terms of a graph Lagrangian. The result is mainly a conceptual bridge between activation-space geometry and classical graph theory.

**Strengths**:
* The main derivation is structured clearly and I did not notice any mathematical inconsistencies.
* The interval-graph perspective is conceptually neat and makes the combinatorial structure of the folding measure more explicit.

**Weaknesses**:
* The ML motivation is not fully clear to me. It’s not obvious why we should care about deviations from discrete convexity along straight input segments, or what properties of learned representations this measure is intended to capture. While these questions may have been addressed in the cited work (particularly the space folding paper by Lewandowski et al.), I think this paper would benefit from a clearer exposition about the applications of the derived results.
* Although the Motzkin-Straus formulation provides a continuous relaxation, the paper does not clearly explain what we would want to optimize.

**Overall assessment**: While the graph-theoretic connection is interesting, strengthening the motivation and clarifying what this perspective enables in studying neural networks beyond reinterpretation of existing theory would make the contribution more compelling.

**Pmlr Suitability:**

NA

---

### Official Review · Reviewer_ofQm · 2026-02-23

**Rating:** 6
**Confidence:** 4

**Review:**

In this paper authors use the space folding measure to traces geometry of neural network's activation. Some comments to consider in revision:

Overall a simple, useful and theoretically justified result on tracking of geometry of representations. Theory seems sound and is cleanly supported with examples.

According to Lemma 4.1 ordering can be arbitrary $D_i$ and $I_i$, which means different orderings can produce different graphs $(G*)$ with different Lagrangians $\Lambda(G*)$, even when $\chi$ is preserved. It is unclear whether the representation is unique or what the implications of non-uniqueness are.

Lack of clarity in what vertices and edges represent in the graph. It appears edges in $G*$ depend solely on temporal overlap of mismatch blocks not any architectural relationship. A single neuron $j$ can contribute multiple disjoint mismatch blocks if it flips on and off several times along the walk. In that case, each block becomes a separate vertex which means the vertices of $G*$ do not correspond to neurons. Graph is just an artifact of one walk and it is unclear if it encodes any neural representations.

Motzkin--Straus theorem is simply used as a change of variables which is trivial. Authors themselves accept that one can compute the same quantity with a trivial O(L) scan so the whole graph construction seems redundant for the case analyzed in the paper. It is unclear if there is any other value.

**Pmlr Suitability:**

NA

---

### Meta-Review · Area_Chair_pqJy · 2026-02-27

**Decision:**

Accept

**Metareview:**

This paper is largely conceptual. It shows that space folding in NNs can be analyzed using the Motzkin–Straus graph Lagrangian. The program has appeared in ML adjacent areas from time to time, such as in https://arxiv.org/abs/2305.08519. The main motivation of the paper is via limitations in existing "space folding measures." Such measures evaluate how NNs map convex regions in the input space to folded non-convex structures in the activation space. The Motzkin-Straus connection is made to translate discrete, step-based activation walks into a continuous optimization landscape via the graph Lagrangian. The main issue is that the paper is purely conceptual, and affords no computational advantage. The utility is unclear. But given that this is a tiny paper, I think it might be useful to be presented at the workshop in case it is able to facilitate more discussion.

**Relevance To Proceedings:**

Tiny paper — does not apply

**Relevance To Workshop:**

Yes — suitable for GRaM

---

### Decision · Program_Chairs · 2026-03-02

Accept (Poster)